# Machine learning of multimodal MRI to predict the development of epileptic seizures after traumatic brain injury

**Marianna La Rocca** [1]                                       MARIANNA.LAROCCA@LONI.USC.EDU
**Rachael Garner** [1]                                          RACHAEL.GARNER@LONI.USC.EDU
**Kay Jann** [1]                                                KAY.JANN@LONI.USC.EDU
**Hosung Kim** [1]                                              HOSUNG.KIM@LONI.USC.EDU
**Paul Vespa** [2]                                              PVESPA@MEDNET.UCLA.EDU
**Arthur W Toga** [1]                                           TOGA@LONI.USC.EDU
**Dominique Duncan** [1]                                        DOMINIQUE.DUNCAN@LONI.USC.EDU

[1] *Laboratory of Neuro Imaging, Stevens Neuroimaging and Informatics Institute, Keck School of Medicine, University of Southern California, Los Angeles, CA, USA.*

[2] *David Geffen School of Medicine at UCLA, Los Angeles, CA, USA.*

## Abstract

One common consequence of Traumatic Brain Injury (TBI) that causes significant disability amongst patient populations is post-traumatic epilepsy (PTE). In this work, we use machine learning approaches to reveal subtle brain changes in Magnetic Resonance Images (MRIs) which may serve as PTE biomarkers.

**Keywords:** Machine learning, Random Forest, Support Vector Machine, FreeSurfer, Multi-Scale Entropy, Magnetic Resonance Imaging, Post-traumatic epilepsy.

## 1. Introduction

PTE develops in approximately $15 - 20\%$ of patients with severe head trauma and is diagnosed if one or more unprovoked seizures occur at least one week after the TBI (Diaz-Arrastia et al., 2009). The Epilepsy Bioinformatics Study for Antiepileptogenic Therapy (EpiBioS4Rx) has dedicated significant effort to identify biomarkers of epileptogenesis that may help prevent seizure occurrence and better understand the mechanisms underlying PTE, which remain unclear (Duncan et al., 2018; Verellen and Cavazos, 2010). Recent studies have demonstrated that machine learning approaches allow for individual-level detection of pathology (Amoroso et al., 2018; La Rocca et al., 2018). In this work, we trained two machine learning systems, Random Forest (RF) and Support Vector Machine (SVM), with morphological features extracted from structural MRI (sMRI) and Multi-Scale Entropy (MSE) measures extracted from resting-state functional MRI (rs-fMRI), with the goal to predict which EpiBioS4Rx patients experienced at least one seizure after TBI. Structural features consist of gray matter (GM) and white matter (WM) volumes by which alterations related to epilepsy may be captured. Functional features are based on MSE, a novel nonlinear statistical metric that can be used for fMRI time series to assess the complexity of the blood oxygen level-dependent (BOLD) signal at each voxel (Costa et al., 2002) and may reflect some aspects of altered local excitatory-inhibitory balance that underlie epilepsy (Wang et al., 2018). To validate the potential of the novel MSE metric in the domain of

the seizure prediction after TBI, we compared MSE features with morphological features extracted by using FreeSurfer (FS). We evaluated the prediction power of these multimodal features and we examined which anatomical areas, according to RF prediction, are proved to be statistically associated to seizure occurrence.

## 2. Material and Methods

57 structural T1 MRI scans of TBI subjects collected as part of EpiBioS4Rx were used. For 30 subjects, the rs-fMRI scan was acquired and thus used to compute the MSE measures. The subjects had a mean age and standard deviation of $38.36 \pm 20.11$, at the time of injury, and a mean Glasgow Coma Scale and standard deviation of $9.96 \pm 4.07$, at the time of the MRI. Among these subjects, 19 experienced at least one seizure after TBI. sMRI scans, acquired according to the MPRAGE protocol, were processed with FS v.6.0. The FS pipeline automatically performs all steps necessary to compute morphological features (Fischl, 2012). This tool allowed us to obtain 188 features for each MRI scan, including subcortical and cortical gray matter volumes, white matter volumes, total gray and white matter volumes, and intracranial volume. rs-fMRI preprocessing included motion realignment and correction of physiological noise. Then, MSE maps were coregistered to individual anatomical T1 images and normalized to the MNI 152 template where the Harvard-Oxford atlas is defined. For each of the binarized cortical and subcortical regions of interest (ROIs) of the atlas, we averaged MSE and generated 69 ROI-based MSE features (48 cortical and 21 subcortical GM regions). We evaluated and compared the prediction power of the FS and MSE features with two classifiers: RF and SVM. For each classifier and for each feature group, we performed 100 rounds of 5-fold cross-validation stratified according to the clinic. For each round, a nested feature selection was carried out with RF to select the most important features in terms of mean accuracy decrease. Finally, we examined, for each feature group, the most important anatomical areas selected over all the cross-validation rounds by picking the features whose importance was greater than 85th quantile of the normalized importance distribution.

## 3. Results

Table 1: Classification performances for FS and MSE features. Performances, assessed with RF and SVM, are reported in terms of accuracy, specificity, sensitivity, and Area Under the receiver operating characteristic Curve (AUC) along with the relative standard deviations.

| Learning model | Accuracy | Specificity | Sensitivity | AUC |
|---|---|---|---|---|
| RF with FS features | $0.67 \pm 0.03$ | $0.61 \pm 0.05$ | $0.71 \pm 0.04$ | $0.71 \pm 0.03$ |
| SVM with FS features | $0.65 \pm 0.03$ | $0.66 \pm 0.04$ | $0.64 \pm 0.04$ | $0.69 \pm 0.04$ |
| RF with MSE features | $0.70 \pm 0.05$ | $0.70 \pm 0.05$ | $0.70 \pm 0.06$ | $0.73 \pm 0.04$ |
| SVM with MSE features | $0.63 \pm 0.04$ | $0.60 \pm 0.04$ | $0.67 \pm 0.04$ | $0.67 \pm 0.04$ |

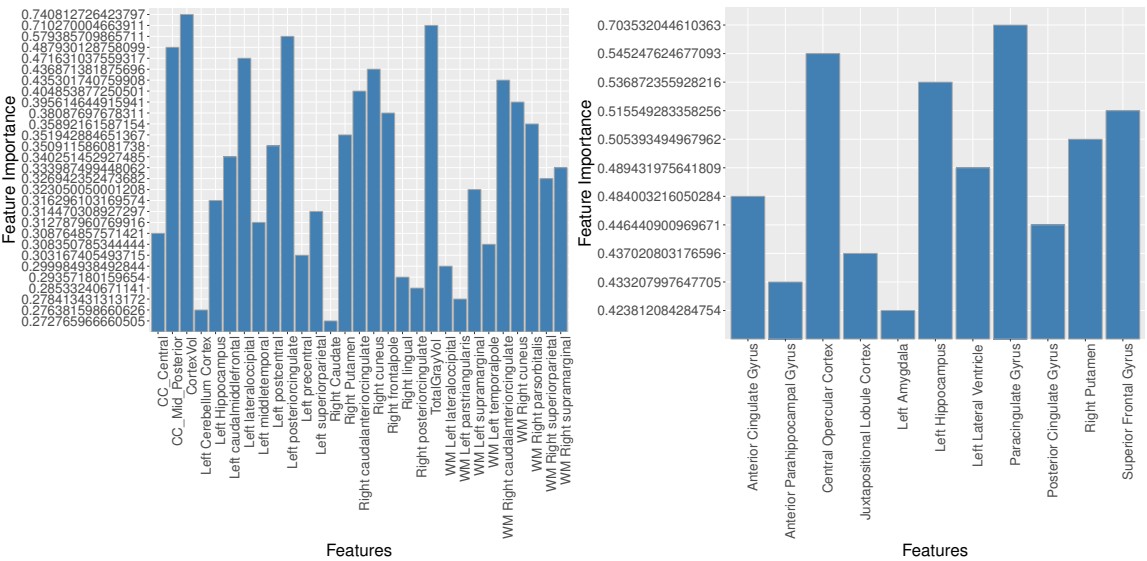

Figure 1: Bar plot of the importance of the most predictive FS features (on the left) and the most predictive MSE features (on the right). Features were selected over all the rounds of cross-validation and each feature corresponds to an anatomical region.

Classification performances obtained with the two machine learning models, RF and SVM, are reported in Table 1 for FS and MSE features. In Figure 1, important features obtained with RF are reported for both FS and MSE features.

## 4. Conclusions

We have shown that FS and MSE features are useful to distinguish TBI patients who develop seizures from those who do not. Best classification performances were achieved using RF, suggesting the complexity of the proposed imaging features. Most of the important regions, such as left hippocampus, right putamen and cingulate gyrus, are accordant for both sets of structural and functional features, suggesting that the TBI-induced structural alterations may further derive abnormal neuronal activity related to seizure generation. We are currently acquiring more samples, which will allow the evaluation of the statistical power when the two modal features are combined in multivariate fashion.

### Acknowledgments

This study was conducted with the support of the National Institute of Neurological Disorders and Stroke (NINDS) of the National Institutes of Health (NIH) under award number U54 NS100064 (EpiBioS4Rx).

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
