# OpenReview forum: "Machine learning of multimodal MRI to predict the development of epileptic seizures after traumatic brain injury"
_MIDL.io/2019/Conference/Abstract — MIDL Abstract 2019_

### Official Review · AnonReviewer2 · 2019-04-30
**unclear explanation to calculate MSE features**

**Rating:** 2
**Confidence:** 2

**Review:**

The authors aimed to identify features associated with seizures after Traumatic Brain Injury (TBI), and proposed novel features as MSE features. The features were used for Support Vector Machine (SVM) and Random Forest (RF) for classification. Then the trained classifiers gave the most associated features based on feature importance.

Some points are unclear:
1) motivation of MSE features, as the authors concluded that both FS and MSE are useful for distinguishing seizure after TBI, does this indicate that using both of them reveal more features for the prediction or are they equivalently useful?
2) calculation of MSE features, the authors explained this mainly in sec. 2 , however it is unclear how MSE measures are computed from rs-fMRI;
3) Classification labels, SVM and RF are used to perform classification, labels for training are not explicitly described, is it patient with and without seizures?

---

### Official Review · AnonReviewer1 · 2019-05-01
**interesting preliminary study; would benefit from further analysis**

**Rating:** 3
**Confidence:** 2

**Review:**

This study seeks to address an important clinical question (predicting the development of seizures following traumatic brain injury). I find this exploration to be interesting, though further comparisons would be helpful for future work:
* As MSE is not a commonly used feature in rsfMRI, it would be useful (as a baseline) to compare this metric to other, more typical metrics computed from fMRI time series (e.g. network / correlation-based measures).
* since MSE may be sensitive to motion/outliers, further temporal pre-processing would likely be helpful.
* it was also not clear to me what is the current "state of the art" for this problem - what kind of accuracy levels have been obtained in relevant prior work, and which features have been previously examined? - or is this the first such study? Therefore, it's hard to assess the degree to which the present results reflect an improvement over existing work.

---

### Decision · Program_Chairs · 2019-05-06
**Acceptance Decision**

Accept